

# Ground subsidence effects on simulating dynamic high latitude surface inundation under permafrost thaw using CLM5

**Altug Ekici[1,2,3], Hanna Lee[1], David M Lawrence[4], Sean C Swenson[4], and Catherine Prigent[5]**

[1]NORCE Norwegian Research Centre, Bjerknes Centre for Climate Research, Bergen, Norway
[2]Climate and Environmental Physics, Physics Institute, University of Bern, Bern, Switzerland
[3]Oeschger Centre for Climate Change Research, University of Bern, Bern, Switzerland
[4]Climate and Global Dynamics Division, National Center for Atmospheric Research, Boulder, Colorado, USA
[5]LERMA, Observatoire de Paris, PSL Research University, CNRS, UMR 8112, F-75014, Paris, France

Correspondence to: *ekici@climate.unibe.ch*

## Abstract

Simulating surface inundation is particularly challenging for the high latitude permafrost regions. Ice-rich permafrost thaw can create expanding thermokarst lakes as well as shrinking large wetlands. Such processes can have major biogeochemical implications and feedbacks to the climate system by altering the pathways and rates of permafrost carbon release. However, the processes associated with it have not yet been properly represented in Earth system models. We show a new model parameterization that allows direct representation of surface water dynamics in CLM (Community Land Model), the land surface model of several Earth System Models. Specifically, we coupled permafrost-thaw induced ground subsidence and surface microtopography distribution to represent surface water dynamics in the high latitudes. Our results show increased surface water fractions around western Siberian plains and northeastern territories of Canada. Additionally, localized drainage events correspond well to severe ground subsidence events. Our parameterization is one of the first steps towards a process-oriented representation of surface hydrology, which is crucial to assess the biogeochemical feedbacks between land and the atmosphere under changing climate.

## 1. Introduction

Northern high latitudes experience pronounced warming due to Arctic amplification (Serreze and Francis, 2006). Within the last decades, temperature increase in the Arctic has been twice the amount of that in the tropics (Solomon et al., 2007). The abrupt increase in Arctic temperatures threatens to destabilize the global permafrost areas and can alter land surface structures, which can lead to releasing considerable amounts of permafrost carbon as greenhouse gases to the climate system (Schuur et al., 2008). The balance between $CO_2$ and $CH_4$ release from permafrost depends largely on the organic matter decomposition pathway; larger inundated areas release more $CH_4$ than $CO_2$ using the anaerobic pathway but overall release of greenhouse gases is greater under aerobic conditions (Lee et al. 2014; Treat et al. 2015). The main natural sources of $CH_4$ emissions are from tropical wetlands, however the contributions from high latitude wetlands are increasing each decade (Saunois et al., 2016) with further thawing of permafrost.



With high percentage of surface wetland coverage (Grosse et al., 2013; Muster et
al., 2017), characterizing high latitude $CH_4$ emissions require detailed process
representations in models. However, Earth system models (ESMs) used in the
future climate projections struggle to represent the complex
physical/hydrological changes in the permafrost covered high latitude regions.
Therefore, it is necessary to improve model representation of surface hydrology
processes within the ESMs.
Permafrost processes have now been represented commonly within the land
surface models (Lawrence et al., 2008; Gouttevin et al., 2012; Ekici et al., 2014;
Chadburn et al., 2015), however, the complex hydrological feedbacks between
degrading permafrost and thermokarst lake formations have been a major
challenge. An extensive review of wetland modeling activities and an
intercomparison effort of evaluating methane-modeling approaches are given in
Wania et al. (2013) and Melton et al. (2013). These studies, however, do not
include permafrost specific features such as excess ice in frozen soils, therefore
they have tendency to under-represent key processes associated to permafrost
thaw. Excess ice melt within the frozen soils can lead to abrupt changes in the
surface topography, creating subsided ground levels, which can enhance pond
formation often recognized as thermokarst formation. Such changes in surface
microtopography can be very effective in altering the soil thermal and
hydrological conditions (Zona et al., 2011).
Lee et al. (2014) implemented surface subsidence processes in the Community
Land Model (CLM: Oleson et al., 2013; Lawrence et al., 2011; Swenson et al.,
2012) to overcome some of the limitations in representing processes associated
with permafrost thaw and subsequent land surface subsidence. The surface
conditions altered by the subsidence events change the microtopography of the
area, which can further modify the surface hydrological conditions in reality. Lee
et al. (2014) did not further couple the land surface subsidence with hydrological
processes to represent subsequent changes in local hydrology created under
permafrost thawing. Here we developed a conceptual coupling of excess ice
melting and subsequent land surface subsidence with hydrology and show how
implementing permafrost thaw induced subsidence affects surface
microtopography distribution and surface inundation in the CLM model.
**2. Methods**
Simulating the effects of permafrost thaw on surface water dynamics requires a
complex interaction of thermodynamics and hydrology within the model. Here
we use the 1° spatial resolution simulations of CLM5 (Lawrence et al., submitted
2018) to represent such dynamics. CLM is a complex, process based terrestrial
ecosystem model simulating biogeophysical and biogeochemical processes
within the soil and vegetation level. Lee et al. (2014) have presented the excess
ice implementation into CLM. The ground excess ice data from International
Circum-Arctic Map of Permafrost and Ground-Ice Conditions (Brown et al., 1997)
are used to create an initial soil ice dataset to be prescribed into the model. The
excess ice in the model undergoes physical phase change but most importantly
melting ice allows a first-order estimation of land surface subsidence under
permafrost thaw.





In CLM, surface inundated fraction ($f_{h2osfc}$) of each grid cell is calculated by using
the microtopography distribution ($\sigma_{micro}$) and the surface water level ($d$) of the
grid cell (Eq. 1 - 3).

$$f_{h2osfc} = \frac{1}{2}\left(1 + erf\left(\frac{d}{\sigma_{micro}\sqrt{2}}\right)\right)$$

Eq.1: Parameterization of surface inundated fraction '$f_{h2osfc}$' using an error function of
surface water level '$d$' (height in m relative to the gridcell mean elevation) and
microtopography distribution '$\sigma_{micro}$' (m).

$$\sigma_{micro} = \left(\beta + \beta_0\right)^{\eta}$$

Eq. 2: Microtopography distribution '$\sigma_{micro}$' as a function of slope, where $\beta$ is the
prescribed topographic slope.
$$\beta_0 = \left(\sigma_{max}\right)^{\frac{1}{\eta}}$$

Eq. 3: Adjustable coefficient $\beta_0$ as a function of maximum topographical distribution
'$\sigma_{max}$'. Original value for $\sigma_{max}$ is 0.4 while $\eta$ is -3.

This parameterization is similar to the TOPMODEL approach (Beven and Kirkby,
1979), where a hypsometric function is used to define the height of standing
water ($d$) within the gridbox by assuming a normal statistical distribution of
ground level microtopography. In this study, the subsidence levels from
permafrost thaw induced excess ice melt are coupled with $\sigma_{micro}$ in order to
represent the naturally occurring subsided landscapes within the permafrost-
affected areas. With increasing excess ice melt, more subsidence occurs and the
amount of subsidence redefines the surface $\sigma_{micro}$, which is inversely related to
$f_{h2osfc}$ (Eq. 1). Therefore, to represent increased $f_{h2osfc}$, $\sigma_{micro}$ has to be decreased
in value. However, $\sigma_{micro}$ is the statistical distribution of surface
microtopography, hence cannot be directly related to physical subsidence levels.
Therefore, a preliminary method of relating $\sigma_{micro}$ to an order of magnitude lower
ground subsidence levels is used (Eq. 4).

$$\sigma'_{micro} = \begin{cases} \sigma_{micro} - s \div b, s < 0.5 \\ \sigma_{micro} + s \div b, s \geq 0.5 \end{cases}$$

Eq. 4: New microsigma parameterization '$\sigma'_{micro}$' where '$s$' is the subsidence in meters
and '$b$' is the adjustable parameter set to 10.

We implemented a conditional formulation regarding the severity of subsidence.
In general, the surface is forced to allow more ponding of water with moderate
levels of subsidence. However, advance levels of excess ice melt can degrade the
surface levels so much that the small troughs created from the initial degradation
can connect to create a drainage system that the grid box can no longer support





any ponding (Liljedahl et al., 2016). For this reason, the excess ice melt has a
reversed effect on $\sigma_{micro}$ after a threshold value of 0.5 m (Eq.4). Choice of this
threshold value is discussed in the following section.
We performed several experiments using CLM5 to assess the general response of
surface hydrology to changing microsigma parameter values. First, the
dependence of $f_{h2osfc}$ to $\sigma_{micro}$ is investigated by doubling $\sigma_{micro}$ (experiment:
Sigma-2) and reducing it by half (experiment: Sigma-0.5). Afterwards, the new
$\sigma_{micro}$ parameterization (experiment: Exice) is compared to the default model
version (experiment: Control), where subsidence does not alter $\sigma_{micro}$ or $f_{h2osfc}$
and to a satellite driven data product (GIEMS, the Global Inundation Extent from
Multiple Satellites, Prigent et al., 2012). All experiments include 155-year
transient simulations following a spin up procedure of repeating 1901-1930
climate forcing for 100 years. The transient 155-year simulation represents the
time period from 1860 till 2015. CRU-NCEP (Viovy, 2009), a combined dataset of
Climate Research Unit (CRU) and National Center for Environmental Protection
(NCEP) reanalysis datasets, is used as the atmospheric forcing for these
experiments.

The GIEMS surface inundation dataset from Prigent et al. (2007, 2012) is used to
compare the simulated inundated fractions. GIEMS uses a combination of
satellite observations to derive the distribution and dynamics of the global
surface water extent. The inundated areas are calculated using passive
microwave observations from Special Sensor Microwave/Imager (SSM/I),
active microwave observations from the scatterometer on board the European
Remote Sensing (ERS) satellite and the normalized difference vegetation index
(NDVI) from the Advanced Very High resolution Radiometer (AVHRR). The
dataset provides monthly-mean values of surface water area from 1993 to
2007, with a spatial resolution of 0.25°. The dataset is spatially projected onto a
1° resolution grid for comparison with the model results.

**32  3. Results and Discussion**
In our experiments, surface inundation ($f_{h2osfc}$) increases where surface
microtopography distribution ($\sigma_{micro}$) decreases (Fig. 1) as expected from the
CLM parameterization. When $\sigma_{micro}$ decreases (Sigma-0.5) compared to the
original value (shown in Supplementary Figure S1), it results in very high $f_{h2osfc}$
over western Siberia and Hudson Bay area, while increasing $\sigma_{micro}$ (Sigma-2)
results in lower $f_{h2osfc}$ in general. In the original CLM parameterization, $f_{h2osfc}$ is
calculated with a static microtopography index (Fig. S1) derived from a
prescribed topographic slope dataset (Oleson et al., 2013).

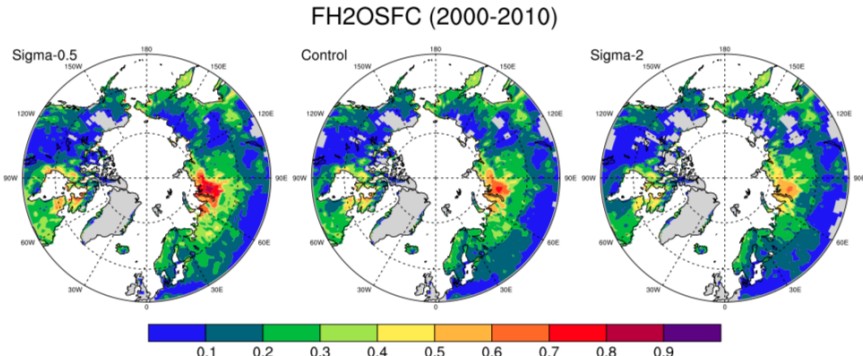

Fig. 1: High latitude (>50°N) maps of simulated surface water fractions ($f_{h2osfc}$) from
Control, Sigma-0.5, and Sigma-2.0 experiments with different microsigma distributions
averaged for the period 2000-2010.
Our results illustrate the dependence of $f_{h2osfc}$ on $\sigma_{micro}$ and how certain range of
$\sigma_{micro}$ values can result in very high $f_{h2osfc}$. This relation emphasize the need for a
dynamic circum-Arctic $\sigma_{micro}$ value to capture the natural variability of surface
conditions when representing permafrost thaw associated hydrological changes.
In the Exice experiment, coupling excess ice melt induced ground subsidence to
$\sigma_{micro}$ leads to significant changes in surface hydrology (Fig. 2). In our
simulations, $\sigma_{micro}$ is consistently lower in Exice compared to Control at the end
of the 20th century (Fig. 2a). This is the model representation of increased
variability in surface microtopography due to uneven subsidence events within
the gridcell. Particularly larger inundated fractions are simulated around
western Siberia and northeast Canada, which conform well to the observational
datasets of peatland distribution (Tarnocai et al., 2007; 2009). Several other
observational estimates agree on the spatial distribution of high latitude
peatlands, where most of the wetland formations are expected in the future
(Melton et al., 2013). Therefore, the new parameterization of surface inundated
fraction is a stepping-stone towards a more realistic representation of surface
hydrology in permafrost-affected areas. Other modeling studies support these
results with similar spatial patterns of surface wetland distributions (Wania et
al., 2013; Melton et al., 2013). In the previous version of CLM, simulated
inundated area shows slightly different patterns (Riley et al., 2011), mainly due
to non-process based description of inundated fractions. We emphasize that
although our parameterization is only conceptual, this is the first attempt
towards coupling permafrost thaw associated land surface subsidence with
hydrological changes in a land surface model within an ESM.
By introducing the effects of ground subsidence on $\sigma_{micro}$, a dynamic inundated
fraction is calculated. However, there is no observed dataset to evaluate the
relation between subsidence and ground topography, therefore an assumption
had to be made regarding this coupling. In this study, changes in $\sigma_{micro}$ are
proportional to the changes in ground subsidence with the difference in an order
of magnitude. This assumption is put to test by doubling and halving the initial
$\sigma_{micro}$ values and the results show 10 to 20 % change in surface inundated
fractions (Fig. 1). The difference in dynamic parameterization (Fig. 2b) stays in



between these values and on average shows a 10 – 15 % increase, thus
supporting the coupling assumption.

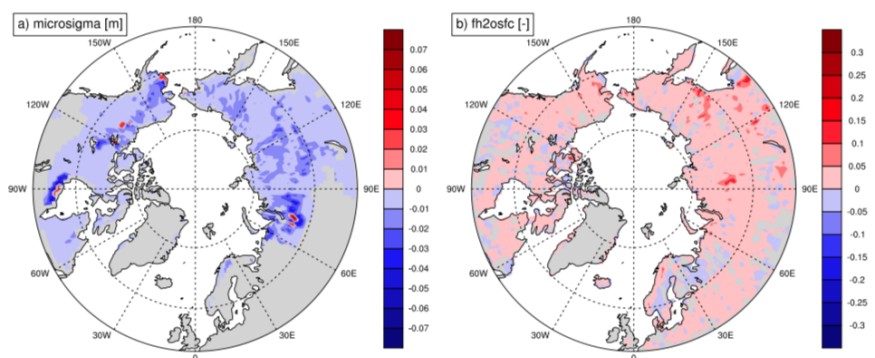

Fig. 2: Effects of coupled subsidence-microsigma parameterization on '$\sigma_{micro}$' and '$f_{h2osfc}$'
from >50°N difference maps of Exice-Control experiments for the period 2000-2010.
As expected, the $f_{h2osfc}$ and $\sigma_{micro}$ changes are directly related to the ground
subsidence processes in most cases. Exice experiment produces land surface
subsidence in some gridcells (Fig. 3) similar to the spatial patterns exhibited in
$\sigma_{micro}$ and $f_{h2osfc}$ in Fig. 2, suggesting that melting of excess ice directly affects
changes in surface hydrology. This is most pronounced around western Siberia,
south of Hudson Bay and around northwestern Canada and central Alaska,
where initial excess ice was large (Lee et al. 2014). Simulated ground subsidence
is directly associated to changes in surface inundated fraction ($f_{h2osfc}$) described
in Fig. 2.
As a result of subsidence threshold parameterization (see Methods), reversed
effect of excess ice melting is shown in the $\sigma_{micro}$ plots (Fig. 2a), where red points
are directly related to the severe ground subsidence locations (Fig. 3). These
areas consistently exhibit abrupt melting of excess ice leading to increased $\sigma_{micro}$.
Larger negative deviations of $\sigma_{micro}$ from the original values were observed in
central Alaska, northwestern Canada, south of Hudson Bay, southwest Russia,
central Siberia, and northern Yakutia regions of Russia (areas with dark blue in
Fig2a). In reality, different landscapes should have a different threshold value,
yet our work is aimed to capture the overall changes and general patterns rather
than local conditions, so a preliminary choice of a single threshold value is used.
Same areas show increased $f_{h2osfc}$ compared to Control (Fig. 2b). The largest
increases in $f_{h2osfc}$ are observed in central Siberia and southeastern Russia, while
some minor decreases in $f_{h2osfc}$ values are present in an unevenly distributed
pattern. It is important to add that the choice of 0.5 m threshold is arbitrary and
can be modified according to the surface dataset of excess ice.





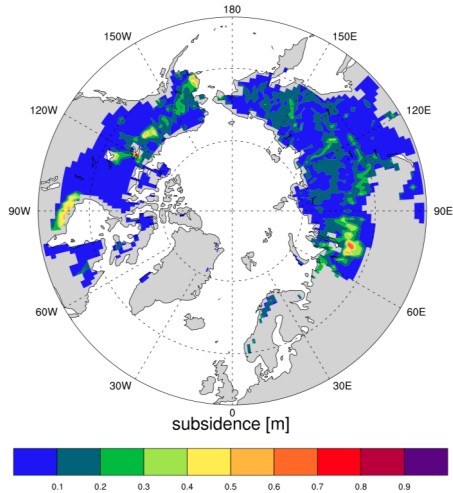

Fig. 3: High latitude (>50°N) map of ground subsidence simulated from the Exice
experiment averaged for the period 2000-2010.
Spatially averaged timeseries of $\sigma_{\mathrm{micro}}$ and $f_{\mathrm{h2osfc}}$ show that in the Exice
experiment $\sigma_{\mathrm{micro}}$ decreases over time and $f_{\mathrm{h2osfc}}$ shows a more dynamic change
during the simulation (Fig. 4). The discrepancy in $\sigma_{\mathrm{micro}}$ between Exice and
Control in the beginning of the simulation is due to prior excess ice melting
during the spin-up period and the values continue to decrease throughout the
20th century, while the decrease halts temporarily during 1960- 1990
(microsigma-diff plot in Fig. 4). Higher $f_{\mathrm{h2osfc}}$ are observed in Exice experiment,
however, the differences between Exice and Control show a general increase
throughout the simulation except the period between 1960-1990. The spatially
averaged $f_{\mathrm{h2osfc}}$ values exhibit a non-linear progression during the 20th century
(Fig. 4). Mainly the change in climate forcing contributes to this trend. Analyzing
the CRUNCEP atmospheric forcing data suggests that the precipitation pattern
over the experiment domain shows a sudden reduction at the beginning of 1960s
(Fig. S2). Even though the average precipitation starts increasing again, the
lower values contribute to the reduced $f_{\mathrm{h2osfc}}$ values. Similar changes occur with
the patterns in atmospheric temperatures (Fig. S2), which is a direct forcing for
permafrost thaw and ground subsidence. A process-based representation of
$f_{\mathrm{h2osfc}}$ allows the model to naturally represent the temporal changes in climate.
Hence, our representation of $f_{\mathrm{h2osfc}}$ will improve the estimation of future surface
hydrological states under changing climatic conditions.



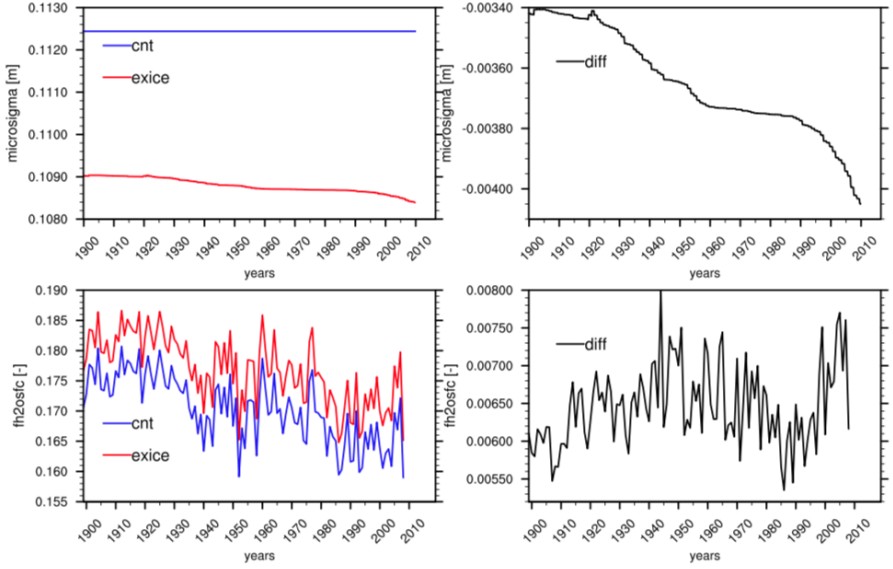

Fig. 4: Timeseries of spatially averaged high latitude (>50°N) $\sigma_{micro}$ and annual
maximum $f_{h2osfc}$ variables from Exice and Control experiments together with the
timeseries of Exice-Control difference (diff) for the period 1900-2010.
The direct effects of new model parameterization can better analyzed while
inspecting point scale changes as shown in Fig. 5. The three selected points show
a range of scenarios to observe the effects of subsidence on microsigma and
$f_{h2osfc}$. Point 1 has no change in subsidence during the simulation and with
higher microsigma values in Exice (due to prior subsidence in spinup), the
difference in $f_{h2osfc}$ compared to Control simulation is always positive, meaning
higher surface inundated fractions. In Point 2, Exice microsigma decreases due to
the increase in subsidence during the simulation. These gradual changes are
reflected in $f_{h2osfc}$, where sudden increases are shown around 1935 and 1955,
exactly when the subsidence changes occur. Similarly in Point 3, subsidence
causes a lower microsigma in the beginning of the simulation; however the
subsidence values surpass the 0.5m threshold around 1920s, which causes the
reversed effect on microsigma by increasing it compared to the Control
experiment. Severe subsidence causing more drainage is represented in this way
within our parameterization. The $f_{h2osfc}$ values show this drainage with a sudden
decrease at 1920 and continuing with mostly negative values throughout the
simulation. These scenarios support the validity of our new parameterization
that can be used for any future climate scenario for a better representation of
surface hydrology and subsidence coupling.





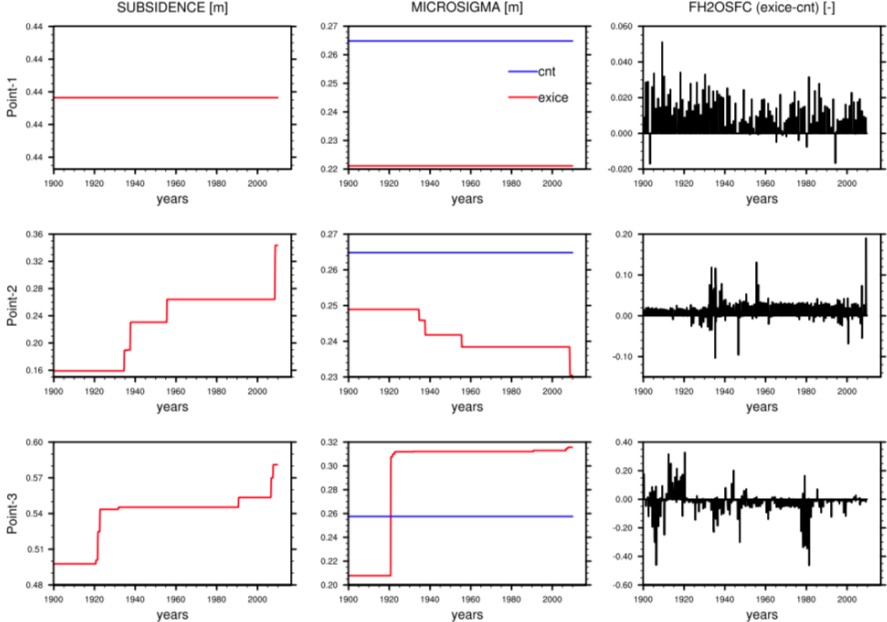

Fig. 5: Timeseries of subsidence, $\sigma_{\text{micro}}$, and $f_{\text{h2osfc}}$ variables from Exice and Control
experiments at three selected sites. Point 1: lat 54 N lon 272 E, Point 2: lat 64 N lon 80 E,
Point 3: lat 65 N lon 70 E.
GIEMS dataset (Prigent et al., 2012) provides the surface area of wetlands for each
gridbox. Fraction of wetland-covered gridbox is calculated to compare with the model
results (Fig. 6). The range of estimated surface wetland fraction is different in the
satellite dataset and model outputs; however, spatial distribution of surface inundated
area is fairly comparable between the model and the satellite dataset. They both
exhibit larger inundated fractions in western Siberia and around Hudson Bay. The
ranges of estimated surface wetland fraction between the satellite dataset and
model outputs are different due to differences in the definitions of inundated
areas. However, spatial distribution of surface inundated area is comparable
between the model and the satellite dataset, where both exhibit larger inundated
fractions in western Siberia and Hudson Bay. Since our model provides the
fraction of gridbox that is inundated, the satellite dataset had to be converted
from actual wetland area to fractions. The GIEMS dataset assumes 773 km²
gridboxes all over the globe (Prigent et al., 2007), which creates grid-size
problems comparing to model gridbox area.   Another issue with such
comparison stems from the differences in the definition of inundated fraction.
GIEMS dataset uses satellite observations at different wavelengths to derive the
wetland area, while the CLM creates the surface inundation with the topography
index and water inputs to the gridbox. Within the model parameterization, the
height of the surface water level is calculated by a hypsometric function and the
gridbox fraction is further derived from the grid size. This allows an ever-
existing surface inundated fraction even in very dry gridboxes, whereas the
GIEMS method underestimates the small wetlands comprising less than 10% of
the gridbox area (Prigent et al., 2007); hence a model overestimation of satellite



dataset is expected. Definition of modelled and satellite derived inundated
fraction is not the same. Unfortunately there is no standard definition
(Reichhardt, 1995), which produces the struggle to find a proper observational
dataset to evaluate model results. What we emphasize from our findings is,
nevertheless, the spatial patterns of higher inundated fractions occurring at
similar locations in model and satellite dataset (Fig. 6).
Fig. 6: Surface water fraction comparison from high latitude (>50°N) maps of annual
maximum surface wetlands from GIEMS dataset (Prigent et al., 2012) and annual
maximum $f_{h2osfc}$ values of Exice and Control experiments for the period 1993-2007.
**4. Conclusion**
A warming climate affects the Arctic more severely than the rest of the globe.
Increasing surface temperatures pose an important threat to the vulnerable high
latitude ecosystems. Degradation of Arctic permafrost due to increased soil
temperatures leads to the release of permafrost carbon to the atmosphere and
further strengthens the greenhouse warming (IPCC, 2013; Schuur et al., 2008).
For future climate predictions, it is necessary to properly simulate the Arctic
surface inundated areas due to their physical and biogeochemical coupling with
the atmosphere.

This study summarizes a new parameterization within the CLM to represent
prognostic surface inundated fractions under permafrost thawing using a
conceptual approach that can lead to implementation of a physical process-based
parameterization. Coupling ground subsidence to surface microtopography
distribution, hence allowing a natural link between surface hydrological
conditions and soil thermodynamics, resulted in generally increased surface
inundated fractions over the northern high latitudes, with larger surface
inundated fractions around western and far-east Siberian plains and
northeastern Canada. Projected increase in global temperatures will inevitably
cause more excess ice melting and subsequent ground subsidence, therefore, it
will be necessary to incorporate a process-based parameterization to accurately
account for future ground subsidence effects on surface hydrological states.

Our results confirm the enhancements of coupling ground subsidence and
surface inundation to represent the temporal changes in surface hydrology





reflected by soil physical states and the atmospheric forcing, which is much
needed for a future scenario experiment. Here we conclude that our new
parameterization is implemented successfully and can be used for future climate
scenarios such as shown in Lee et al. (2014) with major subsidence events
during the 21st century under a high warming scenario.
This new parameterization represents the first step into a process-based
representation of such hydrological processes in CLM. Using this
parameterization, further work can proceed to investigate the biogeochemical
feedbacks of permafrost greenhouse gas fluxes between land and atmosphere.
**Code and data availability**
The code modifications to CLM model in accordance to this paper are accessible
through the Zenodo archive with the following link:
https://zenodo.org/badge/latestdoi/183611414
The overall CLM model code can be obtained from the NCAR archives, the
instructions on accessing the model code is given through this website:
http://www.cesm.ucar.edu/models/cesm2/land/
The full set of model data will be made publicly available through the Norwegian
Research Data Archive at https://archive.norstore.no upon publication.
**Author contribution**
AE and HL designed the experiments and AE carried them out. DML and SCS
developed the main CLM model code and HL developed the previous version this
model is based on. CP has provided the GIEMS dataset. AE performed the
simulations and prepared the manuscript with contributions from all co-authors.

**Acknowledgements**
This work was supported by the Research Council of Norway projects
PERMANOR (255331) and MOCABORS (255061) and NSF EaSM-L02170157. The
simulations were performed on resources provided by UNINETT Sigma2-the
National Infrastructure for High Performance Computing and Data Storage in
Norway, accounts NS2345K and NN2345K.





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
