# Peer review of "Ground subsidence effects on simulating dynamic high latitude surface inundation under permafrost thaw using CLM5"

_Geoscientific Model Development, 2019_

## Referee Comment (RC1) · Anonymous Referee #1 · 31 May 2019

General comments:

Ekici et al. investigated the effects of permafrost thaw induced ground subsidence on terrestrial hydrology modeled in CLM5. The proposed parameterization is not mechanistic, but it is a necessary step towards representing permafrost thaw induced changes in land surface property and hydrology in an Earth System Model (ESM). The paper is well written and it demonstrates an important yet missing element in contemporary ESMs.

Specific comments:

Pg1, L43: In addition to increased temperature, the projected increases in high latitude

precipitation could also accelerate the release of permafrost carbon (e.g., Chang et al., 2019; Grant et al., 2017).

Pg1, L46-47: Can you give a quantitative description about the release of greenhouse gases (e.g., in terms of g CO2-eq/m2)? How about Knoblauch et al. (2018) that found strong CH4 production under anoxic conditions?

Pg2, L1-3: There are many other "detailed processes representations" that can alter high latitude CH4 emissions in addition to surface wetland coverage. For example, the representations of permafrost thaw stage, surface topography, vegetation and microbial community compositions (e.g., Grant et al., 2017; Malhotra & Roulet, 2015; McCalley et al., 2014; Olefeldt et al., 2013).

Pg7 Fig. 3: It might be a good idea to include the simulated soil temperature map here to (1) confirm it aligns reasonably with the simulated ground subsidence; (2) give a sense of how much warming leads to this amount of ground subsidence. Also, if the blue regions (subsidence<0.1m) are close to 0 degree C, wouldn't it suggest a potentially strong ground subsidence with the projected warming after 2010?

Pg8, Fig.4: The spatially averaged sigma-micro between the two sets of runs are very similar. Can you include the variability along with the mean values? It appears that the model is extremely sensitive to a parameter (sigma-micro) that exhibits limited temporal variability. How do the author propose to find realistic sigma-micro values for contemporary and future simulations? Once the parameterization proposed in this study is applied to ESMs, it will trigger significant changes in surface hydrology and thereby biogeochemical feedbacks resulting from sigma-micro selection along (not including the parameterization uncertainty).

Reference:

Chang, K.-Y., Riley, W. J., Crill, P. M., Grant, R. F., Rich, V. I., & Saleska, S. R. (2019). Large carbon cycle sensitivities to climate across a permafrost thaw gradient in subarctic Sweden. The Cryosphere, 13(2), 647–663. https://doi.org/10.5194/tc-13-647-2019

Grant, R. F., Mekonnen, Z. A., Riley, W. J., Arora, B., & Torn, M. S. (2017). Mathematical Modelling of Arctic Polygonal Tundra with Ecosys: 2. Microtopography Determines How CO2 and CH4 Exchange Responds to Changes in Temperature and Precipitation. Journal of Geophysical Research: Biogeosciences, 122(12), 3174–3187. https://doi.org/10.1002/2017JG004037

Knoblauch, C., Beer, C., Liebner, S., Grigoriev, M. N., & Pfeiffer, E. M. (2018). Methane production as key to the greenhouse gas budget of thawing permafrost. Nature Climate Change, pp. 1–4. https://doi.org/10.1038/s41558-018-0095-z

Malhotra, A., & Roulet, N. T. (2015). Environmental correlates of peatland carbon fluxes in a thawing landscape: Do transitional thaw stages matter? Biogeosciences, 12(10), 3119–3130. https://doi.org/10.5194/bg-12-3119-2015

McCalley, C. K., Woodcroft, B. J., Hodgkins, S. B., Wehr, R. A., Kim, E.-H., Mondav, R., et al. (2014). Methane dynamics regulated by microbial community response to permafrost thaw. Nature, 514(7523), 478–481. https://doi.org/10.1038/nature13798

Olefeldt, D., Turetsky, M. R., Crill, P. M., & Mcguire, A. D. (2013). Environmental and physical controls on northern terrestrial methane emissions across permafrost zones. Global Change Biology, 19(2), 589–603. https://doi.org/10.1111/gcb.12071

---

## Referee Comment (RC2) · Anonymous Referee #2 · 11 Jun 2019

**Review:Ground subsidence effects on simulating dynamic high latitude surface inundation under permafrost thaw using CLM5**

**1   General comments**

The fate of carbon stored in soils in the high northern latitudes is determined by landscape features whose horizontal extent is often below the kilometre-scale. Thus, finding an adequate representation of (near-) surface subgrid-scale processes in permafrost affected regions is an issue that is of great importance for modellers working at the coarser scales. Here, Ekici et al. propose a simple parametrization that they use to link the Comunity Land Model's subsidence submodule to the calculation of the inundated fraction by changing the microtopography wherever subsidence occurs.

In the study, the authors show how the inundated fraction reacts to extreme changes in the microtopography, i.e. by halving and doubling the paramter used to describe the microtopography distribution. Furthermore, for selected grid-boxes they demonstrate how the simulated subsidence affects the microtography and the inundated fraction and they show how the surface water fraction in the high northern latitudes differs in simulations with and without their new scheme.

[Figure]

To the best of my knowledge, there is no process-based model representing the lake/wetland dynamics resulting from the melting of ground ice that is suitable for the use in large scale models. Therefore, the proposed parametrizations could very well help to capture the resulting effects in scenario simulations using ESMs. However, even though the article is generally well structured and well written, there are a some key issues that should be addressed prior to publication.

One of my main concerns with the parametrization pertains to the use of the accumulated subsidence for estimating the changes in microtopography. I see this as problematic as subsidence in the model can only increase over time (as the Lee scheme does not account for the formation of soil ice) and the authors introduce a fixed threshold above which further subsidence increases the microtopographic parameter rather than decreasing it. Hence, for the scheme to produce meaningful changes in inundated fraction, it does not only need to be initialized with the correct microtopography and soil ice content but also with reliable information of how much subsidence has happened in the past in any given grid box; in other words one would have to know, how close the subsidence is to passing the threshold when it will lead to an increase in sigma; and I am not aware of any dataset that could provide this information. In their study the authors avoid this initialization problem by starting the simulation with zero-subsidence and then having a 100-year spin-up period. But in this case, the results will be highly dependant on the selection of the spin up period, e.g. if the spin up of the model would have been done for 1000 instead of 100 years the results could look very different as in many grid-boxes the subsidence may have already passed the 0.5m-threshold meaning that the inundated fraction would actually decrease during the simulation.

Additionally, even though the authors make it clear that this is merely a first step, I am not fully convinced by the arguments that are being made in favour of the chosen

parametrisations/assumptions. On pages 5 (l. 31) - 6 (l. 2) the authors claim that the simulated changes in inundated fraction stay within the range that results from halving or doubling the reference value of sigma; but I fail to see how that validates the coupling assumption? It merely shows that the parametrisation has a certain sensitivity, but how sensitive should it actually be? Also, while I can see a certain spatial correlation between the simulated subsidence and the changes in microtopography, i.e. Fig 3 and Fig 2a, I have a very hard time seeing any meaningful correlation between the changes in microtopgraphy (Fig. 2a) and changes in inundated fraction (Fig. 2b). But most importantly, I am not convinced by the comparison to the GIEMS dataset (page 9, l.6 - page 10, l.6). In Figure 6. there is almost no difference in the inundated fractions simulated with the two model versions. And if there was any difference I do not understand how that could demonstrate that it is beneficial to use the new scheme. The control simulation uses the present day sigma and should therefore also result in the best simulated present day inundated fractions. If the simulations with the new scheme give inundated fractions that are closer to the observations (which is not visible in the plots) it merely means that the function CLM uses to compute the inundated fraction could be improved, but not that the reference microtopography is wrong. So at best this comparison shows that the new scheme doesn't change the microtopography so much that it substantially affects the simulated present day inundated fraction. As the scheme is used to capture the dynamics related to subsidence, it would be key to show a comparison with observed trends/changes in the inundated fraction, in order to demonstrate that the scheme performs well.

Consequently, until the authors demonstrate the scheme's ability to improve the models surface water dynamics and provide a strategy for the initialization and spinup of the model, I can not agree with the their conclusion that "the parametrization is implemented successfully and can be used for further climate scenarios".

**2 Specific comments**

- p.2, l.24-l.27: As the subsidence simulated by the scheme is a key input to your model it would be very helpful if you could provide some more details on the scheme by Lee et al..

- P.3, l.32: Why preliminary?

- P.3, l.35: Here, it would be very helpful if you could clarify whether s is indeed the accumulated subsidence since the beginning of the simulation.

- P.4, l.12ff: Is there a specific reason why you do the spinup using the forcing from 1901-1930 while you start your simulation in the year 1860? Wouldn't it make more sense to use the climate forcing from the beginning?.

- P.4, l.18ff: Could you also indicate how the microtopography was initialized in the Exice experiments. I just assumed you use the same index that is used for the control simulation (Fig. S1).

- Fig. 1: I find it quite difficult to judge the differences in fh2osfc between the simulations. Maybe you could show the differences between sigma-0.5 and sigma-2 as a sub-figure? Or maybe you could also provide a graph with sigma and d on the x and y axes and fh2osfc as a colour to give a more systematic overview?

- P.6, l.1f: I fail to see how this supports your coupling assumption. It merely says something about the sensitivity of your parametrization. Without knowing which sensitivity should be expected it is very hard to use this in support for the assumption.

- Fig. 2: I was quite surprised to see so little spatial correlation between the change in microtopography and the change in inundated fraction (could you maybe calculate a correlation coefficient). While sigma is almost exclusively lower in Exise,
there is actually quite a number places where the inundated fraction is also smaller. Additionally, most of the areas in which you find the strongest changes in microtopography show now substantial increase in the inundated fraction. Thus I would not say that the patterns are similar. Here I think more information, especially on the changes in the surface water level, is required for the reader to better understand the plots.

- P.6, l.8-l.13f: I find this formulation problematic. The connection between melting ground ice and surface hydrology is not suggested by the correlations between Figs2 and 3, but because the connections where directly implemented with Lee et al.'s and your scheme. But, while I do see a correlation between Figs 2a and 3, I do not see the same patterns in Fig 2b.

- P.7, l.7ff: If you initialize your simulation with the present day sigma and the present day ice content, and then run it for 240 years (spinup + 1860 - 2000) during which time the ice content can only decrease, wouldn't you necessary end up with a worse microtopography for present day?. I presume that the initialisation/ spinup procedure was carried out because there is no data to consistently initialize the model either at 1860 or at present day? But what would be the strategy to initialize/ spin up the model for future simulations?

- Fig. 4 and Fig 5.: Why is the difference in fh2osfc so variable even if there are no pronounced changes in sigma and the two experiments use the same forcing?.

---

## Referee Comment (RC3) · Anonymous Referee #3 · 20 Jun 2019

General comments:

Ekici et al. propose a new model parameterization to represent surface water dynamics caused by ground subsidence in CLM (Community Land Model). The subsidence level is coupled with a microtopography parameter in TOPMODEL approach. This study is the first step to quantify complicated processes in permafrost regions with Earth system models.

Special comments:

p.3 l.1: Cound you explain the effect of the modified parameters (e.g. microtpography distribution and surface inundated fraction) on the entire model? Those descriotions

would be helpful to understand the proposed parameterization is crucial to assess the biogeochemical feedbacks.

p.3 l.24-33: The delineation of the actual relathinship between ground subsidence and microtopography is necessary to understand the relevance of modeling instead of a required parameterization by governing equations in CLM.

p.3 l.35: Related to the previous comment, if you could calculate more realistic value of microsigma with finer-resolution topographic data and subsidence information, does it improve the model applicability? It would be helpful if you explain the limitation of "modeling (conceptulization)" and "parameterization" respectively.

p.10 fig.6: As the authors pointed out, it is difficult to directly compare inundated area between GIEMS dataset and simulated results due to the gap of definitions of water surface. However, I think some other variables relating water budget (e.g. river discharge) are modified by the proposed parameterization and can be compared with observation data. I apologize if I misunderstand the numerical implementation in CLM.
* * *

---

## Referee Comment (RC4) · Anonymous Referee #4 · 25 Jun 2019

This is an interesting study about a new parameterization of surface water dynamics including the effect of ground subsidence induced by excess ice melt in high latitudes. A simulation with the new parameterization shows increased surface water fractions in most areas compared to the control simulation. Though the purpose of the study is solid and sound, the manuscript has some major issues that need to be addressed before final publication.

Major issues:

1. The assumption for the proposed equation may not be correct. The authors assumed that decreased micro topography distribution (microsigma) represents in-

creased surface inundation (fh2osfc). However, from equation 1, it seems that the assumption is true only when surface water level d is greater than 0, hence fh2osfc is greater than 0.5.

2. When the subsidence value surpasses the 0.5 m threshold, microsigma suddenly increases as shown in the lower middle panel of figure 5. Does this sudden change represent any physical processes?

3. Is there any feedback from surface water fraction to other hydrology variables (e.g. soil moisture) in the model?

4. The initial conditions seem to be important for the simulations. How was the spin up period determined? Is 100-years enough for spin up?

5. There are not enough validations performed. It is not clear for me which simulation is better when comparing with the observation. Is it possible to validate time series of surface water fraction using GIEMS dataset?

Minor issues:

P1 L28-31: Is this true? The authors say "The largest increases in fh2osfc are observed in central Siberia and southeastern Russia" on line 27 of page 6.

P3 Equation 2: What is eta?

P5 L11-15: This is confusing. sigma_micro is defined as microtopography distribution on line 9 of page 3. Why does lower sigma_micro represent increased variability in surface microtopography?

P8 L10: "higher microsigma" should be "lower microsigma"?

---

## Author Comment (AC1) · 9 Aug 2019

**Response to reviewer comment 1:**

Specific comments

Pg1, L43: In addition to increased temperature, the projected increases in high latitude precipitation could also accelerate the release of permafrost carbon (e.g., Chang et al., 2019; Grant et al., 2017).

- We thank the reviewer for the suggested references. These new references will be added in the revised manuscript.

Pg1, L46-47: Can you give a quantitative description about the release of greenhouse gases (e.g., in terms of g CO2-eq/m2)? How about Knoblauch et al. (2018) that found strong CH4 production under anoxic conditions?

- We will add the following section in the revised manuscript version:

"However, for a future model estimate, Knoblauch et al (2018) predicts twice as much permafrost carbon release in anoxic conditions (241±138 g CO2 kgC-1) compared to oxic conditions (113±58 g CO2 kgC-1) by 2100.".

Pg2, L1-3: There are many other "detailed processes representations" that can alter high latitude CH4 emissions in addition to surface wetland coverage. For example, the representations of permafrost thaw stage, surface topography, vegetation and microbial community compositions (e.g., Grant et al., 2017; Malhotra & Roulet, 2015; McCalley et al., 2014; Olefeldt et al., 2013).

- We thank the reviewer for these suggestions. We agree to include more detailed processes that influence the high latitude CH4 emissions in the revised version. The following section will be added in the revised manuscript:

"Besides surface wetland conditions, models should also properly estimate permafrost thaw stage (Malhotra & Roulet, 2015), changing surface topography (Olefeldt et al., 2013), and surface vegetation and microbial conditions (Grant et al., 2017) in order to improve estimations of surface CH4 emissions."

Pg7 Fig. 3: It might be a good idea to include the simulated soil temperature map here to (1) confirm it aligns reasonably with the simulated ground subsidence; (2) give a sense of how much warming leads to this amount of ground subsidence. Also, if the blue regions (subsidence<0.1m) are close to 0 degree C, wouldn't it suggest a potentially strong ground subsidence with the projected warming after 2010?

- We thank the reviewer for this suggestion and we want to emphasize that the scope of our current work is the connection between subsidence and surface water since the relation between subsidence and soil temperature/moisture was thoroughly discussed in the previous work: Lee et al. 2014. So for the sake of keeping the manuscript concise, we would like to refer to Lee et al. (2014) for the soil temperature diagnostics.

- The blue regions with subsidence <0.1m, the reviewer mention here, can indeed indicate a strong subsidence in the future where the soil temperatures are close to 0. We would like to emphasize that this is one of the motivations

to use our new parameterization for future simulations and investigate the subsidence under warming scenarios.

Pg8, Fig.4: The spatially averaged sigma-micro between the two sets of runs are very similar. Can you include the variability along with the mean values? It appears that the model is extremely sensitive to a parameter (sigma-micro) that exhibits limited temporal variability. How do the author propose to find realistic sigma-micro values for contemporary and future simulations? Once the parameterization proposed in this study is applied to ESMs, it will trigger significant changes in surface hydrology and thereby biogeochemical feedbacks resulting from sigma-micro selection along (not including the parameterization uncertainty).

- We understand the reviewer's concern about the strength of microsigma parameter in our model. The variability of spatially averaged microsigma in Exice experiment is quite small indeed (variance: 2.8e-8, standard dev.:1.6e-4), so for the figure it doesn't make sense to add these in the manuscript. With the current knowledge, there is no perfect way to optimize the microsigma parameter for each gridbox in global simulations, this is why we tried to estimate micro-sigma by coupling to other well-known physical processes like excess ice melt. Since there is no global dataset to directly compare with our model results, one should be cautious interpreting our model's contemporary and future estimates. One avenue to constrain our parameterization will be to use the terrestrial greenhouse gas fluxes, once we use the biogeochemistry coupled to our parameterization, and this is for the next step in our work.

---

## Author Comment (AC2) · 9 Aug 2019

**Response to reviewer comment 2:**

General Comments

One of my main concerns with the parametrization pertains to the use of the accumulated subsidence for estimating the changes in microtopography. I see this as problematic as subsidence in the model can only increase over time (as the Lee scheme does not account for the formation of soil ice) and the authors introduce a fixed threshold above which further subsidence increases the microtopographic parameter rather than decreasing it. Hence, for the scheme to produce meaningful changes in inundated fraction, it does not only need to be initialized with the correct microtopography and soil ice content but also with reliable information of how much subsidence has happened in the past in any given grid box; in other words one would have to know, how close the subsidence is to passing the threshold when it will lead to an increase in sigma; and I am not aware of any dataset that could provide this information. In their study the authors avoid this initialization problem by starting the simulation with zero-subsidence and then having a 100-year spin-up period. But in this case, the results will be highly dependant on the selection of the spin up period, e.g. if the spin up of the model would have been done for 1000 instead of 100 years the results could look very different as in many grid-boxes the subsidence may have already passed the 0.5m-threshold meaning that the inundated fraction would actually decrease during the simulation.

- We acknowledge the reviewer's concern. We agree that this is one of the largest sources of uncertainty in our work. As the reviewer pointed out, our parameterization depends much on the initialization of excess ice and there is currently no global scale dataset to parameterize and evaluate the model. One feasible proxy for evaluating the surface inundation is to use the terrestrial CO2 and CH4 fluxes once we use our parameterization coupled to the CLM biogeochemistry module. This is the aim for the next step in our work and we hope that our work can motivate the observation community to collect such dataset.

- The spin up procedure was sufficiently long enough to bring the physical state into equilibrium, since we did not use the biogeochemistry, we did not need a longer spin up period than 100 years. Also the excess ice melt comes to an equilibrium with the spin up climate state so a longer spin up would not change the initial excess ice melt conditions.

Additionally, even though the authors make it clear that this is merely a first step, I am not fully convinced by the arguments that are being made in favour of the chosen parametrisations/assumptions. On pages 5 (l. 31) - 6 (l. 2) the authors claim that the simulated changes in inundated fraction stay within the range that results from halving or doubling the reference value of sigma; but I fail to see how that validates the coupling assumption? It merely shows that the parametrisation has a certain sensitivity, but how sensitive should it actually be?

- We have chosen to double and halve the reference microsigma value in the sensitivity analysis to show the upper and lower boundary of the sensitivity in fh2osfc with changing microsigma. The behavior of fh2osfc in these sensitivity simulations support that the dynamic parameterization in this study does not

lead to unrealistic fh2osfc values in the simulations under present day climate. This test is merely to constrain any extreme sensitivity cases that might have originated from our conceptual scheme. Finding the best sensitivity of surface inundation to soil subsidence is beyond the scope of this study and currently very challenging to estimate with global observational datasets.

Also, while I can see a certain spatial correlation between the simulated subsidence and the changes in microtopography, i.e. Fig 3 and Fig 2a, I have a very hard time seeing any meaningful correlation between the changes in microtopgraphy (Fig. 2a) and changes in inundated fraction (Fig. 2b).

- The subsidence directly dictates the microsigma changes in the code, therefore, it is more straightforward to diagnose the relation between subsidence and microsigma than subsidence and fh2osfc. We agree with the reviewer that it is difficult to tease out direct relationship between microsigma changes and surface inundation. This is due to the fact that surface inundation is not only affected by the subsidence but also by combination of factors such as precipitation, air temperature, and soil moisture. Hence, the fh2osfc changes in Fig2b is difficult to interpret only from the changes in microtopography under excess ice melting. Yet, we would like to draw the reviewer's attention to the extreme subsidence areas (red points in Fig2a) and the corresponding changes (even though very small) in the surface inundation map (small blue areas inFig 2b), which suggests that our parameterization is creating surface inundation at the areas where it should. Figures 4 and 5 are added for similar reasons to compare the changes in fh2osfc in global and point scale dynamics. The future simulations under climate warming will show pronounced subsidence (Lee et al., 2014) and the consequent effects on surface inundation will be more visible.

But most importantly, I am not convinced by the comparison to the GIEMS dataset (page 9, l.6 - page 10, l.6). In Figure 6. there is almost no difference in the inundated fractions simulated with the two model versions. And if there was any difference I do not understand how that could demonstrate that it is beneficial to use the new scheme. The control simulation uses the present day sigma and should therefore also result in the best simulated present day inundated fractions. If the simulations with the new scheme give inundated fractions that are closer to the observations (which is not visible in the plots) it merely means that the function CLM uses to compute the inundated fraction could be improved, but not that the reference microtopography is wrong. So at best this comparison shows that the new scheme doesn't change the microtopography so much that it substantially affects the simulated present day inundated fraction. As the scheme is used to capture the dynamics related to subsidence, it would be key to show a comparison with observed trends/changes in the inundated fraction, in order to demonstrate that the scheme performs well.

- We thank the reviewer for opening this point to discussion. Fig. 6 indeed does not show a large difference between the Control and Exice simulations. However, as we pointed out in the discussion, we intended to show that our new parameterization does not create unrealistic values compared to the Control simulation and this work is merely to increase our confidence to use the new dynamic parameterization for future climate change scenarios, where

the differences due to major subsidence will be more pronounced. So, we do not claim the current CLM microsigma parameter is faulty, our new parameterization introduces a temporal variability to the microsigma parameter and it shouldn't diverge too much with the present day conditions. Hence, the similarity between Control and Exice simulations in Fig 6 supports our aim. On the other hand, we use the GIEMS dataset to additionally show that the regions where extensive high surface inundation occurs in observational dataset and to confirm that the model results correspond well with the observations in the spatial patterns of surface inundation. Since the GIEMS dataset was not a very long time series, we couldn't use this dataset for direct comparison over time. However, Fig 5 demonstrates model's behaviour in time for different climatic conditions and the deviations from the control run are quite distinguishable.

Consequently, until the authors demonstrate the scheme's ability to improve the models surface water dynamics and provide a strategy for the initialization and spinup of the model, I can not agree with the their conclusion that "the parametrization is implemented successfully and can be used for further climate scenarios".

- We believe we have answered some of the reviewer's concerns and we are not sure if the reviewer has some other suggestions at this point. We want to clarify that one of the points of this manuscript is to show a new parameterization that works globally for a land surface scheme. We suggest to revise our conclusion points to tone down the implications of this study that it is the first step in this kind of parameterization. But more importantly, this study really brings out the importance of observational data and we encourage observations to take this into account.

Specific Comments
- p.2, l.24-l.27: As the subsidence simulated by the scheme is a key input to your model it would be very helpful if you could provide some more details on the scheme by Lee et al..
- we are adding some details of Lee et al. scheme in the methods section in the revised manuscript.
- P.3, l.32: Why preliminary?
- wrong choice of word, changed 'preliminary' to 'conceptual'
- P.3, l.35: Here, it would be very helpful if you could clarify whether s is indeed the accumulated subsidence since the beginning of the simulation.
- yes we added clarification in the text
- P.4, l.12ff: Is there a specific reason why you do the spinup using the forcing from 1901-1930 while you start your simulation in the year 1860? Wouldn't it make more sense to use the climate forcing from the beginning?.
- it was just a standard procedure for CLM to use the 1901-1930 block for the spinup and we wanted to stay consistent.
- P.4, l.18ff: Could you also indicate how the microtopography was initialized in the Exice experiments. I just assumed you use the same index that is used for the control simulation (Fig. S1).

- yes it was using the same reference microsigma. This information is now added in the text
- Fig. 1: I find it quite difficult to judge the differences in fh2osfc between the simulations. Maybe you could show the differences between sigma-0.5 and sigma-2 as a sub-figure? Or maybe you could also provide a graph with sigma and d on the x and y axes and fh2osfc as a colour to give a more systematic overview?
- we are adding the difference map sigma-0.5 - sigma-2 in the supplements

[Figure]

- P.6, l.1f: I fail to see how this supports your coupling assumption. It merely says something about the sensitivity of your parametrization. Without knowing which sensitivity should be expected it is very hard to use this in support for the assumption.
- discussed this above in the main points
- Fig. 2: I was quite surprised to see so little spatial correlation between the change in microtopography and the change in inundated fraction (could you maybe calculate a correlation coefficient). While sigma is almost exclusively lower in Exise, there is actually quite a number places where the inundated fraction is also smaller. Additionally, most of the areas in which you find the strongest changes in microtopography show now substantial increase in the inundated fraction. Thus I would not say that the patterns are similar. Here I think more information, especially on the changes in the surface water level, is required for the reader to better understand the plots.
- this point is also discussed above in the main points
- P.6, l.8-l.13f: I find this formulation problematic. The connection between melting ground ice and surface hydrology is not suggested by the correlations between Figs2 and 3, but because the connections where directly implemented with Lee et al.'s and your scheme. But, while I do see a correlation between Figs 2a and 3, I do not see the same patterns in Fig 2b.

- this point is also discussed above in the main points
- P.7, l.7ff: If you initialize your simulation with the present day sigma and the present day ice content, and then run it for 240 years (spinup + 1860 - 2000) during which time the ice content can only decrease, wouldn't you necessary end up with a worse microtopography for present day?. I presume that the initialisation/ spinup procedure was carried out because there is no data to consistently initialize the model either at 1860 or at present day? But what would be the strategy to initialize/ spin up the model for future simulations?
- yes it is true that the microtopography is expected to be different in accordance to the subsidence levels occurred during the spin up and transient simulation, but the idea here is to constrain the dynamic parameterization and to avoid any major extreme sensitivity from the conceptual method. since there is no way to properly initialize the soil subsidence, we will use other biogeochemical variables (co2/ch4 fluxes) to constrain the surface inundation in our future work, but it is out of scope of this merely model development manuscript.
- Fig. 4 and Fig 5.: Why is the difference in fh2osfc so variable even if there are no pronounced changes in sigma and the two experiments use the same forcing?.
- In the CLM, fh2osfc is also affected by soil and atmospheric changes, however, Fig 5 shows that the changes in microsigma influence fh2osfc on a point scale. This change is difficult to point out in larger spatial scale as in Fig 4, where the spatial averages are used.

---

## Author Comment (AC3) · 9 Aug 2019

**Response to reviewer comment 3:**

Special comments

p.3 l.1: Cound you explain the effect of the modified parameters (e.g. microtpography distribution and surface inundated fraction) on the entire model? Those descriotions would be helpful to understand the proposed parameterization is crucial to assess the biogeochemical feedbacks.

- We thank the reviewer for this suggestion and the following text will be added to the revised manuscript:

"Surface water is defined by spatial scale elevation variations that is the microtopography. The microtopography is normally distributed around the grid cell mean elevation. The fractional area of the grid cell that is inundated (fh2osfc) can be calculated with the standard deviation of this microtopographic distribution. The surface inundated fraction, in turn, affects the soil heat/water/carbon fluxes with the atmosphere."

p.3 l.24-33: The delineation of the actual relathinship between ground subsidence and microtopography is necessary to understand the relevance of modeling instead of a required parameterization by governing equations in CLM.

- We are not sure if the reviewer is requesting us to show the relationship between ground subsidence and microtopography in reality, which is hard to assess due to the lack of observational data. As a result, we used existing parameterization in the CLM surface hydrology based on TOP model. We acknowledge that this is only the first step in this kind of parameterization and hope that our study can bring attention to observational community for such observational data.

p.3 l.35: Related to the previous comment, if you could calculate more realistic value of microsigma with finer-resolution topographic data and subsidence information, does it improve the model applicability? It would be helpful if you explain the limitation of "modeling (conceptulization)" and "parameterization" respectively.

- The parameterization in models such as CLM should focus more on functionality and that this is a very conceptual step in the parameterization. Next step will be subgrid-scale representation of this process but this is not within the scope of our study. We refer to Aas et al. (2019) for the subgrid scale process representation in the revised manuscript.

p.10 fig.6: As the authors pointed out, it is difficult to directly compare inundated area between GIEMS dataset and simulated results due to the gap of definitions of water surface. However, I think some other variables relating water budget (e.g. river discharge) are modified by the proposed parameterization and can be compared with observation data. I apologize if I misunderstand the numerical implementation in CLM.

- We thank the reviewer for the question. It is correct that other water budget variables are affected from our parameterization, however for river discharge, the direct effects from the surface subsidence are minor compared to permafrost thaw related spring river discharge increases, hence not useful to validate the new model.

References:

Aas, Kjetil S., et al. "Thaw processes in ice-rich permafrost landscapes represented with laterally coupled tiles in a land surface model." *The Cryosphere* 13.2 (2019): 591-609.

---

## Author Comment (AC4) · 9 Aug 2019

**Response to reviewer comment 4:**

Major issues

1. The assumption for the proposed equation may not be correct. The authors assumed that decreased micro topography distribution (microsigma) represents increased surface inundation (fh2osfc). However, from equation 1, it seems that the assumption is true only when surface water level d is greater than 0, hence fh2osfc is greater than 0.5.

- We are not sure if the reviewer is asking about negative water level conditions, but the model does not allow negative surface water levels (d), so the Eq. 1 does indeed show an inverse relation between microsigma and fh2osfc. We hope this clarifies the reviewer's concern.

2. When the subsidence value surpasses the 0.5 m threshold, microsigma suddenly increases as shown in the lower middle panel of figure 5. Does this sudden change represent any physical processes?

- Yes it does actually. The increase in microsigma represents the extreme cases where soil subsidence leads to drying of the surface (Liljedahl et al. 2016), which is explained in the text (P3: last paragraph, P6: last paragraph).

3. Is there any feedback from surface water fraction to other hydrology variables (e.g. soil moisture) in the model?

- The surface water fraction does affect the evapotranspiration and soil moisture but the effects are small when compared to other factors such as precipitation and permafrost thaw. So, yes, there are feedbacks between surface water fraction and other hydrological variables, but not big enough compared to our new parameterization.

4. The initial conditions seem to be important for the simulations. How was the spin up period determined? Is 100-years enough for spin up?

- The spinup period of 100 years is chosen to make sure the soil physical variables (soil temperature/moisture) are equilibrated. And it was long enough to avoid any big drifts in these variables. However, for the soil subsidence it is hard to determine an optimum spin up period since the contemporary or past soil excess ice data is very uncertain and hard to constrain the initial conditions. However, the soil excess ice melt also stabilizes with the spin up climate so the spin up period was long enough to have an initial condition for our simulations.

5. There are not enough validations performed. It is not clear for me which simulation is better when comparing with the observation. Is it possible to validate time series of surface water fraction using GIEMS dataset?

- Unfortunately it was not possible to use GIEMS dataset to compare the temporal dynamics, since the dataset period was too short to see any major

differences. Also as discussed in response to other reviewers' comments, this paper merely aims to show that the new parameterization is in line with the current model and does not create extreme conditions. We did not expect big changes compared to the Control simulation, but we do plan to use this for future climate warming scenarios, where higher subsidence levels (Lee et al., 2014) will certainly create more distinguished results to Control simulation.

Minor issues

P1 L28-31: Is this true? The authors say "The largest increases in fh2osfc are observed in central Siberia and southeastern Russia" on line 27 of page 6.

- The text in abstract is about the general increases in fh2osfc, whereas the text in P6:L27 is related to the difference between Exice and Control simulations and more about the direct effects of dynamic parameterization. In the end, Fig6 shows that the higher inundated fractions are actually around western Siberia and around the Hudson Bay.

P3 Equation 2: What is eta?

- "eta" here is just an adjustable parameter.

P5 L11-15: This is confusing. sigma_micro is defined as microtopography distribution on line 9 of page 3. Why does lower sigma_micro represent increased variability in surface microtopography?

- This is related to the distribution of surface microtopography. There is increased variability on surface of gridbox but the distribution of different levels is lowered, hence a lower microsigma.

P8 L10: "higher microsigma" should be "lower microsigma"?

- We thank the reviewer for the correction, the mistake is corrected now.